# Query Complexity of Bayesian Private Learning

**Kuang Xu**
Stanford Graduate School of Business
Stanford, CA 94305, USA
kuangxu@stanford.edu

## Abstract

We study the query complexity of Bayesian Private Learning: a learner wishes to locate a random target within an interval by submitting queries, in the presence of an adversary who observes all of her queries but not the responses. How many queries are necessary and sufficient in order for the learner to accurately estimate the target, while simultaneously concealing the target from the adversary?

Our main result is a query complexity lower bound that is tight up to the first order. We show that if the learner wants to estimate the target within an error of $\epsilon$, while ensuring that no adversary estimator can achieve a constant additive error with probability greater than $1/L$, then the query complexity is on the order of $L \log(1/\epsilon)$ as $\epsilon \to 0$. Our result demonstrates that increased privacy, as captured by $L$, comes at the expense of a *multiplicative* increase in query complexity. The proof builds on Fano's inequality and properties of certain proportional-sampling estimators.

## 1   Introduction

*How to learn, while ensuring that a spying adversary does not learn?* Enabled by rapid advancements in the Internet, surveillance technologies and machine learning, companies and governments alike have become increasingly capable of monitoring the behavior of individuals or competitors, and use such data for inference and prediction. Motivated by these developments, the present paper investigates the extent to which it is possible for a learner to protect her knowledge from an adversary who observes, completely or partially, her actions.

We will approach these questions by studying the query complexity of Bayesian Private Learning, a framework proposed by [17] and [13] to investigate the privacy-efficiency trade-off in sequential learning. Our main result is a tight lower bound on query complexity, showing that there will be a price to pay for the learner in exchange for improved privacy, whose magnitude scales *multiplicatively* with respect to the level of privacy desired. In addition, we provide a family of inference algorithms for the adversary, based on proportional sampling, which is provably effective in estimating the target against any learner who does not employ a large number of queries.

### 1.1   The Model: Bayesian Private Learning

We begin by describing the Bayesian Private Learning model formulated by [17] and [13]. A *learner* is trying to accurately identify the location of a random *target*, $X^*$, up to some constant additive error, $\epsilon$, where $X^*$ is uniformly distributed in the unit interval, $[0, 1)$. The learner gathers information about $X^*$ by submitting $n$ *queries*, $(Q_1, \ldots, Q_n) \in [0, 1)^n$, for some $n \in \mathbb{N}$. For each query, $Q_i$, she receives a binary *response*, indicating the target's location relative to the query: $R_i = \mathbb{I}(X^* \le Q_i), \quad i = 1, 2, \ldots, n$, where $\mathbb{I}(\cdot)$ denotes the indicator function.

The learner submits the queries in a sequential manner, and subsequent queries may depend on previous responses. Once all $n$ queries are submitted, the learner will produce an estimator for the target. The learner's behavior is formally captured by a learner strategy, defined as follows.

**Definition 1.1** (Learner Strategy). *Fix $n \in \mathbb{N}$. Let $Y$ be a uniform random variable over $[0, 1)$, independent from other parts of the system; $Y$ will be referred to as the random seed. A learner strategy, $\phi = (\phi^q, \phi^l)$, consists of two components:*

1. Querying mechanism*: $\phi^q = (\phi_1^q, \dots, \phi_n^q)$, is a sequence of deterministic functions, where $\phi_i^q : [0, 1)^{i-1} \times [0, 1) \to [0, 1)$ takes as input past responses and the random seed, $Y$, and generates the next query, i.e.,* [1]

$$Q_i = \phi_i^q(R^{i-1}, Y), \quad i = 1, \dots, n, \tag{1.1}$$

*where $R^i$ denotes the responses from the first $i$ queries: $R^i = (R_1, \dots, R_i)$, and $R^0 \stackrel{\triangle}{=} \emptyset$.*

2. Estimator*: $\phi^l : [0, 1)^n \times [0, 1) \to [0, 1)$ is a deterministic function that maps all responses, $R^n$, and $Y$ to a point in the unit interval that serves as a "guess" for $X^*$: $\widehat{X} = \phi^l(R^n, Y)$.. The estimator $\widehat{X}$ will be referred to as the learner estimator.*

*We will use $\Phi_n$ to denote the family of learner strategies that submit $n$ queries.*

The first objective of the learner is to accurately estimate the target, as is formalized in the following definition.

**Definition 1.2** ($\epsilon$-Accuracy). *Fix $\epsilon \in (0, 1)$. A learner strategy, $\phi$, is $\epsilon$-accurate, if its estimator approximates the target within an absolutely error of $\epsilon/2$ almost surely, i.e.,*

$$\mathbb{P}\left( \left| \widehat{X} - X^* \right| \le \epsilon/2 \right) = 1, \tag{1.2}$$

*where the probability is with respect to the randomness in the target, $X^*$, and the random seed, $Y$.*

We now introduce the notion of *privacy*: in addition to estimating $X^*$, the learner would like to simultaneously conceal $X^*$ from an eavesdropping adversary. Specifically, there is an adversary who knows the learner's query strategy, and observes all of the queries but not the responses. The adversary then uses the query locations to generate her own *adversary estimator* for $X^*$, denoted by $\widehat{X}^a$, which depends on the queries, $(Q_1, \dots, Q_n)$, and any internal, idiosyncratic randomness.

With the adversary's presence in mind, we define the notion of a private learner strategy.

**Definition 1.3** (($\delta, L$)-Privacy). *Fix $\delta \in (0, 1)$ and $L \in \mathbb{N}$. A learner strategy, $\phi$, is $(\delta, L)$-private if, for any adversary estimator, $\widehat{X}^a$,*

$$\mathbb{P}(|\widehat{X}^a - X^*| \le \delta/2) \le 1/L, \tag{1.3}$$

*where the probability is measured with respect to the randomness in the target, $X^*$, and any randomness employed by the learner strategy and the adversary estimator.* [2]

In particular, if a learner employs a $(\delta, L)$-private strategy, then no adversary estimator can be close to the target within an absolute error of $\delta/2$ with a probability great than $1/L$. Therefore, for any fixed $\delta$, the parameter $L$ can be interpreted as the *level of desired privacy*.

We are now ready to define the main quantity of interest in this paper: query complexity.

**Definition 1.4.** *Fix $\epsilon$ and $\delta$ in $[0, 1]$, and $L \in \mathbb{N}$. The query complexity, $N(\epsilon, \delta, L)$, is the least number of queries needed for an $\epsilon$-accurate learner strategy to be $(\delta, L)$-private:*

$$N(\epsilon, \delta, L) \stackrel{\triangle}{=} \min\{n : \Phi_n \text{ contains a strategy that is both } \epsilon\text{-accurate and } (\delta, L)\text{-private}\}.$$

## 2 Main Result

The main objective of the paper is to understand how $N(\epsilon, \delta, L)$ varies as a function of the input parameters, $\epsilon$, $\delta$ and $L$. Our result will focus on the regime of parameters where[3]

$$0 < \epsilon < \delta/4 \text{ and } \delta < 1/L. \tag{2.1}$$

The following theorem is our main result. The upper bound has appeared in [17] and [13] and is included for completeness; the lower bound is the contribution of the present paper.

**Theorem 2.1** (Query Complexity of Bayesian Private Learning). *Fix $\epsilon$ and $\delta$ in $(0,1)$ and $L \in \mathbb{N}$, such that $\epsilon < \delta/4$ and $\delta < 1/L$. The following is true.*

1. *Upper bound:*
$$N(\epsilon, \delta, L) \leq L\log(1/\epsilon) - L(\log L - 1) - 1. \tag{2.2}$$

2. *Lower bound:*
$$N(\epsilon, \delta, L) \geq L\log(1/\epsilon) - L\log(2/\delta) - 3L\log\log(\delta/\epsilon). \tag{2.3}$$

Both the upper and lower bounds in Theorem 2.1 are constructive, in the sense that there is a concrete learner strategy that achieves the upper bound, and an adversary estimator that forces any learner strategy to employ at least as many queries as that prescribed by the lower bound.

If we apply Theorem 2.1 in the regime where $\delta$ and $L$ stay fixed, while the learner's error tolerance, $\epsilon$, tends to zero, we obtain the following corollary in which the upper and lower bounds on query complexity coincide.

**Corollary 2.2.** *Fix $\delta \in (0,1)$ and $L \in \mathbb{N}$, such that $\delta < 1/L$.[4]*

$$N(\epsilon, \delta, L) \sim L\log(1/\epsilon), \quad as \ \epsilon \to 0. \tag{2.4}$$

Note that the special case of $L = 1$ corresponds to when the learner is not privacy-constrained and aims to solely minimize the number of queries. Theorem 2.1 and Corollary 2.2 thus demonstrate that there is a hefty price to pay in exchange for privacy, as the query complexity depends *multiplicatively* on the level of privacy, $L$.

## 3 Motivation and Related Literature

Bayesian Private Learning is a more general model that contains, as a special case ($L = 1$), the classical problem of sequential learning with binary feedback, with applications in statistics ([11]), information theory ([7]) and optimization ([15]). The Bayesian Private Learning model thus inherits these applications, with the additional feature that, instead of being solely interested in minimizing the number of queries, the decision maker would also like to ensure the privacy of the target. Note that our present model assumes that all responses are noiseless, in contrast to some of the noisy query models in the literature (cf. [10, 1, 15]).

Bayesian Private Learning is a variant of the so-called Private Sequential Learning problem. Both models were formulated in [17] and [13], and the main distinction between the two is that the target is drawn *randomly* in Bayesian Private Learning, while it is chosen in a *worst-case* fashion (against the adversary) in the original Private Sequential Learning model. [17] and [13] establish matching

upper and lower bounds on query complexity for Private Sequential Learning. They also propose the Replicated Bisection algorithm as a learner strategy for the Bayesian variant, but without a matching query complexity lower bound. The present paper closes this gap.

At a higher level, our work is related to a growing body of literature on privacy-preserving mechanisms, in computer science (cf. [5, 9, 6]), operations research (cf. [4, 12]), and statistical learning theory (cf. [2, 8, 16]), but diverges significantly in models and applications. On the methodological front, our proof uses Fano's inequality, a fundamental tool for deriving lower bounds in statistics, information theory, and active learning ([3]).

# 4 The Upper Bound

The next two sections are devoted to the proof of Theorem 2.1. We first prove the query complexity upper bound, and begin by giving an overview of the main ideas. Consider the special case of $L = 1$, where learner is solely interested in finding the target, $X^*$, and not at all concerned with concealing it from the adversary. Here, the problem reduces to the classical setting, where it is well-known that the bisection strategy achieves the optimal query complexity (cf. [15]). The bisection strategy recursively queries the mid-point of the interval which the learner knows to contain $X^*$. For instance, the learner would set $Q_1 = 1/2$, and if the response $R_1 = 0$, then she will know that $X^*$ lies in the interval $[0, 1/2]$, and set $Q_2$ to $1/4$; otherwise, $Q_2$ will be set to $3/4$. This process repeats for $n$ steps. Because the size of the smallest interval known to contain $X^*$ is halved with each additional query, this yields the query complexity

$$N(\epsilon, \delta, 1) = \log(1/\epsilon), \quad \epsilon \in (0, 1). \tag{4.1}$$

Unfortunately, once the level of privacy $L$ increases above 1, the bisection strategy is almost *never* private: it is easy to verify that if the adversary sets $\widehat{X}^a$ to be the learner's last query, $Q_n$, then the target is sure to be within a distance of at most $\epsilon$. That is, the bisection strategy is not $(\delta, L)$-private for any $L > 1$, whenever $\epsilon < \delta/2$. This is hardly surprising: in the quest for efficiency, the bisection strategy submits queries that become progressively *closer* to the target, thus rending its location obvious to the adversary.

Building on the bisection strategy, we arrive at a natural compromise: instead of a single bisection search over the entire unit interval, we could create $L$ identical copies of a bisection search across $L$ disjoint sub-intervals of $[0, 1)$ that are chosen ahead of time, in a manner that makes it impossible to distinguish which search is truly looking for the target. This is the main idea behind the Replicated Bisection strategy, first proposed and analyzed in [13]. We examine this strategy in Section 4, which will yield the query-complexity upper bound, on the order of $L \log(1/\epsilon)$.

We now prove the upper bound, which has appeared in [17] and [13], where the authors also proposed, without a formal proof, the Replicated Bisection learner strategy that achieves $(\delta, L)$-privacy with $L \log(1/\epsilon) - L(\log(L) - 1)$ queries. For completeness, we first review the Replicated Bisection strategy and subsequently give a formal proof of its privacy and accuracy. The main idea behind Replicated Bisection is to create $L$ *identical* copies of a bisection search in a strictly symmetrical manner so that the adversary wouldn't be able to know which one of the $L$ searches is associated with the target. The strategy takes as initial inputs $\epsilon$ and $L$, and proceeds in two phases:

*Phase 1 - Non-adaptive Partitioning*. The learner submits $L - 1$ (non-adaptive) queries: $Q_1 = \frac{1}{L}$, $Q_2 = \frac{2}{L}, \ldots, Q_{L-1} = 1 - \frac{1}{L}$. Adjacent queries are separated by a distant of $1/L$, and together they partition the unit interval into $L$ disjoint sub-intervals of length $1/L$ each. We will refer to the interval $[(i-1)/L, i/L]$ as the *$i$th sub-interval*. Because the queries in this phase are non-adaptive, after the first $L - 1$ queries, while the learner knows which sub-interval contains the target, $X^*$, the adversary has gained no information about $X^*$. We will denote by $\mathcal{I}^*$ the sub-interval that contains $X$.

*Phase 2 - Replicated Bisection*. The second phase further consists of a sequence of $K$ *rounds*, $K = \log\left(\frac{1}{L\epsilon}\right)$. In each round, the learner submits one query in each of the $L$ sub-intervals, and the location of the said query relative to the left end of the sub-interval is the same across all sub-intervals. Crucially, in the $k$th round, the query corresponds to the $k$th step in a bisection search carried out in the sub-interval $\mathcal{I}^*$, which contains the target. The rounds continue until the learner has identified the location of $X^*$ with sufficient accuracy within $\mathcal{I}^*$. The queries outside of $\mathcal{I}^*$ serve only the purpose

of obfuscation by maintaining a strict symmetry. Figure 1 in Section A in the Supplemental Note contains the pseudo-code for Phase 2.

Denote by $Q^*$ the last query that the learner submits in the sub-interval $\mathcal{I}^*$ in Phase 2, and by $R^*$ its response. It follows by construction that either $R^* = 0$ and $X^* \in [Q^* - \epsilon, Q^*)$, or $R^* = 1$ and $X^* \in [Q^*, Q^* + \epsilon)$. Therefore, the learner can produce the estimator by setting $\widehat{X}$ to the mid point of either $[Q^* - \epsilon, Q^*)$ or $[Q^*, Q^* + \epsilon)$, depending on the value of $R^*$, and this guarantees of an additive error of at most $\epsilon/2$. We have thus shown that the Replicated Bisection strategy is $\epsilon$-accurate. The following result shows that it is also private; the proof is given in Section B.1 in the Supplemental Note.

**Proposition 4.1.** *Fix $\epsilon$ and $\delta$ in $(0,1)$ and $L \in \mathbb{N}$, such that $\epsilon < \delta/4$ and $\delta < 1/L$. The Replicated Bisection strategy is $(\delta, L)$-private.*

Finally, we verify the number of queries used by Replicated Bisection: the first phase employs $L - 1$ queries, and the second phase uses $L$ queries per round, across $\log(\frac{1}{L\epsilon})$ rounds, leading to a total of $(L-1) + L\log(\frac{1}{L\epsilon}) = L\log(1/\epsilon) - L(\log L - 1) - 1$ queries. This completes the proof of the query complexity upper bound in Theorem 2.1.

## 5 The Lower Bound

**Main Ideas**. We prove the query complexity lower bound in Theorem 2.1 in this section, which turns out to be significantly more challenging than showing the upper bound. To show that the query complexity is at least, say, $n$, we will have to demonstrate that *none* of the learner strategies using $n - 1$ queries, $\Phi_{n-1}$, can be simultaneously private and accurate. Because the sufficient statistic for the adversary to perform estimation is the posterior distribution of the target given the observed queries, a frontal assault on the problem would require that we characterize the resulting target posterior distribution for *all* strategies, a daunting task given the richness of $\Phi_{n-1}$, which grows rapidly as $n$ increases.

Our proof will indeed take an indirect approach. The key idea is that, instead of allowing the adversary to use the entire posterior distribution of the target, we may restrict her to a seemingly much weaker class of *proportional-sampling estimators*, where the estimator $\widehat{X}^a$ is sampled from a distribution proportional to the empirical density of the queries. A proportional-sampling estimator would, for instance, completely ignore the order in which the queries are submitted, which may contain useful information about the target. We will show that, perhaps surprisingly, the proportional-estimators are so powerful that they leave the learner no option but to use a large number of samples. This forms the core of the lower bound argument. The proof further consists of the following steps.

*1. Discrete Private Learning* (Definition 5.2). We formulate a discrete version of the original problem where both the learner and adversary estimate the discrete index associated with a certain sub-interval that contains the target, instead of the continuous target value. The discrete framework is conceptually clearer, and will allow us to deploy information-theoretic tools with greater ease.

*2. Localized Query Complexity* (Lemma 5.4). Within the discrete version, we prove a localized query complexity result: conditioning on the target being in a coarse sub-interval of $[0, 1)$, any accurate learner still needs to submit a large number of queries *within* the said sub-interval. The main argument hinges on Fano's inequality and a characterization of the conditional entropy of the queries and the target.

*3. Proportional-Sampling Estimator* (Definition 5.5). We use the localized query complexity in the previous step to prove a query complexity lower bound for the discrete version of Bayesian Private Learning (Proposition 5.3). This is accomplished by analyzing the performance of the family of proportional-sampling estimators, where the adversary reports index of a sub-interval that is sampled randomly with probabilities proportional to the number of learner queries each sub-interval contains. We will show that the proportional-sampling estimator will succeed with overwhelming probability whenever an accurate learner strategy submits too few queries, thus obtaining the desired lower bound. In fact, we will prove a more general lower bound, where the learner can make mistakes with a positive probability.

*4. From Discrete to Continuous* (Proposition 5.6). We complete the proof by connecting the discrete version back to the original, continuous problem. Via a reduction argument, we show that the

original query complexity is always bounded from below by its discrete counterpart with some modified learner error parameters, and the final lower bound will be obtained by optimizing over these parameters. The main difficulty in this portion of the proof is due to the fact that an accurate continuous learner estimator is insufficient for generating an accurate discrete estimator that is correct almost surely. We will resolve this problem by carefully bounding the learner's probability of estimation error, and apply the discrete query lower bound developed in the previous step, in which the learner is allowed to make mistakes.

**Discrete Bayesian Private Learning**   We begin by formulating a discrete version of the original problem, where the goal for both the learner and the adversary is to recover a discrete index associated with the target, as opposed to generating a continuous estimator. We first create two nested partitions of the unit interval consisting of equal-length sub-intervals, where one partition is coarser than the other. The objective of the learner is to recover the index associated with the sub-interval containing $X^*$ in the finer partition, whereas that of the adversary is to recover the target's index corresponding to the coarser partition (an easier task!). We consider this discrete formulation because it allows for a simpler analysis using Fano's inequality, setting the stage for the localized query complexity lower bound in the next section.

Formally, fix $s \in (0, 1)$ such that $1/s$ is an integer. Define $M_i(s)$ to be the sub-interval

$$M_s(i) = [(i-1)s, is), \quad i = 1, 2, \ldots, 1/s. \tag{5.1}$$

In particular, the set $\mathcal{M}_s := \{M_s(i) : i = 1, \ldots, 1/s\}$ is a partition of $[0, 1)$ into $1/s$ sub-intervals of length $s$ each. We will refer to $\mathcal{M}_s$ as the $s$-uniform partition. Define $J(s, x) = j$, s.t. $x \in M_s(j)$. That is, $J(s, x)$ denotes the indices of the interval containing $x$ in the $s$-uniform partition. A visualization of the index $J(\cdot, \cdot)$ is given in Figure 2 in Section A of the Supplemental Note.

We now formulate an analogous, and slightly more general, definition of accuracy and privacy for the discrete problem. We will use the super-script $D$ to distinguish them from their counterparts in the original, continuous formulation. Just like the learner strategy in Definition 1.1, a discrete learner strategy, $\phi^D$, is allowed to submit queries at any point along $[0, 1)$, and has access to the random seed, $Y$. The only difference is that, instead of generating a continuous estimator, a discrete learner strategy produces an estimator for the *index* of the sub-interval containing the target in an $\epsilon$-uniform partition, $J(\epsilon, X^*)$.

**Definition 5.1** (($\epsilon, \nu$)-accuracy - Discrete Version). *Fix $\epsilon$ and $\nu \in (0, 1)$. A discrete learner strategy, $\phi^D$, is ($\epsilon, \nu$)-accurate if it produces an estimator, $\widehat{J}$, such that $\mathbb{P}\left(\widehat{J} \neq J(\epsilon, X^*)\right) \leq \nu$.*

Importantly, in contrast to its continuous counterpart in Definition 1.2 where the estimator must satisfy the error criterion with probability one, the discrete learner strategy is allowed to make mistakes up to a probability of $\nu$. The role of adversary is similarly defined in the discrete formulation: upon observing all $n$ queries, the adversary generates an estimator, $\widehat{J}^a$, for the index associated with the sub-interval containing $X^*$ in the (coarser) $\delta$-uniform partition, $J(\delta, X^*)$. The notion of ($\delta, L$)-privacy for a discrete learner strategy is defined in terms of the adversary's (in)ability to estimate the index $J(\delta, X^*)$.

**Definition 5.2** (($\delta, L$)-privacy - Discrete Version). *Fix $\delta \in (0, 1)$ and $L \in \mathbb{N}$. A discrete learner strategy, $\phi^D$, is ($\delta, L$)-private if under any adversary estimator $\widehat{J}^a$, we have that $\mathbb{P}\left(\widehat{J}^a = J(\delta, X^*)\right) \leq 1/L$. We will denote by $\Phi_n^D$ as the family of discrete learner strategies that employ at most $n$ queries.*

We are now ready to define the query complexity of the discrete formulation, as follows:

$$N^D(\epsilon, \nu, \delta, L) = \min\{n : \Phi_n^D \text{ contains a strategy that is both } (\epsilon, \nu)\text{-accurate and } (\delta, L)\text{-private}\}.$$

A main result of this subsection is the following lower bound on $N^D$, which we will convert into one for the original problem in Section 5.

**Proposition 5.3** (Query Complexity Lower Bound for Discrete Learner Strategies). *Fix $\epsilon$, $\nu$ and $\delta$ in $(0, 1)$ and $L \in \mathbb{N}$, such that $\epsilon < \delta < 1/L$. We have that*

$$N^D(\epsilon, \nu, \delta, L) \geq L\left[(1-\nu)\log(\delta/\epsilon) - h(\nu)\right], \tag{5.2}$$

*where $h(p)$ is the Shannon entropy of a Bernoulli random variable with mean $p$: $h(p) = -p\log(p) - (1-p)\log(1-p)$ for $p \in (0, 1)$, and $h(0) = h(1) = 0$.*

**Localized Query Complexity Lower Bound**    We prove Proposition 5.3 in the next two subsections. The first step, accomplished in the present subsection, is to use Fano's inequality to establish a query complexity lower bound localized to a sub-interval: conditional on the target belonging to a sub-interval in the $\delta$-partition, any discrete-learner strategy must devote a non-trivial number of queries in that sub-interval if it wishes to be reasonably accurate.[5]

Fix $n \in \mathbb{N}$, and a learner strategy $\phi^D \in \Phi_n^D$. Because the strategy will submit at most $n$ queries, without loss of generality, we may assume that if the learner wishes to terminate the process after the first $K$ queries, then she will simply set $Q_i$ to 0 for all $i \in \{K + 1, K + 2, \ldots, n\}$, and the responses for those queries will be trivially equal to 0 almost surely. Denote by $\mathcal{Q}^j$ the set of queries that lie within the sub-interval $M_\delta(j)$:

$$\mathcal{Q}^j \triangleq \{Q_i, \ldots, Q_n\} \cap M_\delta(j), \tag{5.3}$$

and by $|\mathcal{Q}^j|$ its cardinality. Denote by $\mathcal{R}^j$ the set of responses for those queries in $\mathcal{Q}^j$. Define $\xi_{j,y}$ to be the learner's (conditional) probability of error:

$$\xi_{j,y} = \mathbb{P}\left(\widehat{J} \neq J(\epsilon, X^*) \,\middle|\, J(\delta, X^*) = j, Y = y\right), \quad j \in \{1, \ldots, 1/\delta\}, \, y \in [0, 1). \tag{5.4}$$

Denote by $\mathcal{E}_{j,y}$ the event:

$$\mathcal{E}_{j,y} = \{J(\delta, X^*) = j, Y = y\}. \tag{5.5}$$

We have the following lemma. The proof is based on Fano's inequality, and is given in Section B.2 of the Supplemental Note.

**Lemma 5.4** (Localized Query Complexity).  *Fix $\epsilon$, $\nu$ and $\delta \in (0, 1)$, $\epsilon < \delta$, and an $(\epsilon, \nu)$-accurate discrete learner strategy. We have that*

$$\mathbb{E}\left(|\mathcal{Q}^j| \,\middle|\, J(\delta, X^*) = j, Y = y\right) \geq (1 - \xi_{j,y}) \log(\delta/\epsilon) - h(\xi_{j,y}), \tag{5.6}$$

*for all $j \in \{1, \ldots, 1/\delta\}$, and $y \in [0, 1)$, and $\sum_{j=1}^{1/\delta} \delta \int_0^1 \xi_{i,y} \, dy \leq \nu$.*

**Proportional-Sampling Estimator**    We now use the local complexity result in Lemma 5.4 to complete the proof of Proposition 5.3. The lemma states that if the target were to lie in a given sub-interval in the $\delta$-uniform partition, then an accurate learner strategy would have to place at least $\log(\delta/\epsilon)$ queries within the said sub-interval *on average*.

**Definition 5.5.**  *A proportional-sampling estimator, $\widehat{J}^a$, is generated according to the distribution:*

$$\mathbb{P}\left(\widehat{J}^a = j\right) = \frac{|\mathcal{Q}^j|}{\sum_{j'=1}^{1/\delta} |\mathcal{Q}^{j'}|}, \quad j = 1, 2, \ldots, 1/\delta. \tag{5.7}$$

*That is, an index is sampled with a probability proportional to the number of queries that fall within the corresponding sub-interval in the $\delta$-uniform partition.*

We next bound the probability of correct estimation when the adversary employs a proportional-sampling estimator: for all $j = 1, 2, \ldots, 1/\delta$, we have that

$$\mathbb{P}\left(\widehat{J}^a = J(\delta, X^*) \,\middle|\, \mathcal{E}_{j,y}\right) = \mathbb{P}\left(\widehat{J}^a = j \,\middle|\, \mathcal{E}_{j,y}\right) = \mathbb{E}\left(\frac{|\mathcal{Q}^j|}{\sum_{j'=1}^{1/\delta} |\mathcal{Q}^{j'}|} \,\middle|\, \mathcal{E}_{j,y}\right)$$

$$\stackrel{(a)}{=} \frac{1}{n} \mathbb{E}\left(|\mathcal{Q}^j| \,\middle|\, \mathcal{E}_{j,y}\right) \stackrel{(b)}{\geq} \frac{1}{n}\left((1 - \xi_{j,y}) \log(\delta/\epsilon) - h(\xi_{j,y})\right), \tag{5.8}$$

where step $(a)$ follows from the fact that $\phi^D \in \Phi_n^D$, and hence $\sum_{j'=1}^{1/\delta} |\mathcal{Q}^{j'}| = n$, and step $(b)$ from Lemma 5.4. Recall that the learner strategy is $(\epsilon, \nu)$-private, and the random seed $Y$ has a probability density of 1 in $[0, 1)$ and zero everywhere else. Since Eq. (5.8) holds for all $j$ and $y$, we can integrate

and obtain the adversary's overall probability of correct estimation:

$$\mathbb{P}\left(\widehat{J}^a = J(\delta, X^*)\right)$$

$$=\sum_{j=1}^{1/\delta}\int_0^1 \mathbb{P}(\mathcal{E}_{j,y})\mathbb{P}\left(\widehat{J}^a = J(\delta, X^*)\,\big|\,\mathcal{E}_{j,y}\right)dy \overset{(a)}{\geq} \frac{1}{n}\sum_{j=1}^{1/\delta}\int_0^1 \delta\left[(1-\xi_{j,y})\log(\delta/\epsilon) - h(\xi_{j,y})\right]dy$$

$$=\frac{1}{n}\left[\left(1 - \sum_{j=1}^{1/\delta}\delta\int_0^1 \xi_{j,y}\,dy\right)\log(\delta/\epsilon) - \sum_{j=1}^{1/\delta}\delta\int_0^1 h(\xi_{j,y})\,dy\right]$$

$$\overset{(b)}{\geq} \frac{1}{n}\left[(1-\nu)\log(\delta/\epsilon) - \sum_{j=1}^{1/\delta}\delta\int_0^1 h(\xi_{j,y})\,dy\right]$$

$$\overset{(c)}{\geq} \frac{1}{n}\left[(1-\nu)\log(\delta/\epsilon) - h\left(\sum_{j=1}^{1/\delta}\delta\int_0^1 \xi_{j,y}\,dy\right)\right] \overset{(d)}{\geq} \frac{1}{n}\left[(1-\nu)\log(\delta/\epsilon) - h(\nu)\right], \qquad (5.9)$$

where step $(a)$ follows from Eq. (5.8), steps $(b)$ and $(d)$ from Lemma 5.4, i.e., $\sum_{j=1}^{1/\delta}\delta\int_0^1 \xi_{i,y}\,dy \leq \nu$. Step $(c)$ is a result of Jensen's inequality and the Bernoulli entropy function $h(\cdot)$'s being concave.

Recall that, in order for a learner strategy to be $(\delta, L)$-private, we must have that $\mathbb{P}\left(\widehat{J}^a = J(\delta, X^*)\right) \leq \frac{1}{L}$ for *any* adversary estimator $\widehat{J}^a$. Eq. (5.9) thus implies that $n \geq L\left[(1-\nu)\log(\delta/\epsilon) - h(\nu)\right]$ is a necessary condition. Because this holds for any accurate and private learner policy, we have thus proven Proposition 5.3.

**From Discrete to Continuous Strategies** We now connect Proposition 5.3 to the original continuous estimation problem. The next proposition is the main result of this subsection. The core of the proof is a reduction that constructs a $(\beta\epsilon, \beta^{-1})$-accurate and $(\delta, L)$-private discrete learner strategy from an $\epsilon$-accurate and $(\delta, L)$-private continuous learner strategy. The proof is given in Section B.3 of the Supplemental Note.

**Proposition 5.6.** *Fix $\epsilon$ and $\delta$ in $(0,1)$ and $L \in \mathbb{N}$, such that $\epsilon < \delta/4$ and $\delta < 1/L$. Fix $\beta \in [2, \delta/\epsilon]$.[6] We have that*

$$N(\epsilon, \delta, L) \geq N^D(\beta\epsilon, \beta^{-1}, \delta, L). \qquad (5.10)$$

**Completing the Proof of the Lower Bound** We are now ready to establish the query complexity lower bound in Theorem 2.1. Fix $\epsilon$ and $\delta$ in $(0,1)$ and $L \in \mathbb{N}$, such that $\epsilon < \delta/4$ and $\delta < 1/L$. Using Propositions 5.3 and 5.6, we have that for any $\beta \in [2, \delta/\epsilon]$,

$$N(\epsilon, \delta, L) \geq N^D(\beta\epsilon, \beta^{-1}, \delta, L) \overset{(b)}{\geq} L\left[(1-\beta^{-1})\log\left(\frac{\delta}{\epsilon}\beta^{-1}\right) - h(\beta^{-1})\right], \qquad (5.11)$$

where the last step follows from Proposition 5.3 by substituting $\nu$ with $\beta^{-1}$ and $\epsilon$ with $\beta\epsilon$. Letting $\gamma \overset{\triangle}{=} \beta^{-1}$, the above inequality can be rearranged to become

$$\frac{N(\epsilon, \delta, L)}{L} \geq (1-\gamma)\log\left(\frac{\delta}{\epsilon}\gamma\right) - h(\gamma) \overset{(a)}{\geq} (1-\gamma)\log\left(\frac{\delta}{\epsilon}\gamma\right) + 2(1-\gamma)\log(\gamma)$$

$$\geq \log(\delta/\epsilon) - \gamma\log(\delta/\epsilon) + 3\log\gamma, \qquad (5.12)$$

where step $(a)$ follows from the assumption that $\gamma = \beta^{-1} \leq 1/2$, and the fact that $h(x) \leq -2(1-x)\log(x)$ for all $x \in (0, 1/2]$. Consider the choice: $\beta = \log(\delta/\epsilon)$. To verify $\beta$ still belongs to the range $[2, \delta/\epsilon]$, note that the assumption that $\epsilon < \delta/4$ ensures $\beta \geq 2$, and because $x > \log(x)$ for all $x > 0$, we have that $\beta < \delta/\epsilon$. Substituting $\gamma$ with $(\log(\delta/\epsilon))^{-1}$ in Eq. (5.12), we have that $\frac{N(\epsilon,\delta,L)}{L} \geq \log(\delta/\epsilon) - 1 - 3\log\log(\delta/\epsilon)$ or, equivalently, $N(\epsilon, \delta, L) \geq L\log(1/\epsilon) - L\log(2/\delta) - 3L\log\log(\delta/\epsilon)$. This completes the proof of the lower bound in Theorem 2.1.

# 6 Concluding Remarks

The main contribution of the present paper is a tight query complexity lower bound for the Bayesian Private Learning problem, which, together with an upper bound in [13], shows that the learner's query complexity depends multiplicatively on the level of privacy, $L$: if an $\epsilon$-accurate learner wishes to ensure that an adversary's probability of making a $\delta$-accurate estimation is at most $1/L$, then she needs to employ on the order of $L \log(\delta/\epsilon)$ queries. Moreover, we show that the multiplicative dependence on $L$ holds even under the more general models of high-dimensional queries and partial adversary monitoring. To prove the lower bound, we develop a set of information-theoretic arguments which involve, as a main ingredient, the analysis of proportional-sampling adversary estimators that exploit the action-information proximity inherent in the learning problem.

The present work leaves open a few interesting directions. Firstly, the current upper and lower bounds are not tight in the regime where the adversary's error criterion, $\delta$, is significantly smaller than $1/L$. Making progress in this regime is likely to require a more delicate argument and possible new tools. Secondly, our query model assumes that the responses are noiseless, and it will be interesting to explore how may the presence of noise (cf. [10, 1, 15]) impact the design of private query strategies. For instance, a natural generalization of the bisection search algorithm to the noisy setting is the Probabilistic Bisection Algorithm ([7, 15]), where the $n$th query point is the median of the target's posterior distribution in the $n$th time slot. It is conceivable that one may construct a probabilistic query strategy analogous to the Replicated Bisection strategy by replicating queries in $L$ pre-determined sub-intervals. However, it appears challenging to prove that such replications preserve privacy, and still more difficult to see how one may obtain a matching query complexity lower bound in the noisy setting. Finally, one may want to consider richer, and potentially more realistic, active learning models, such as one in which each query reveals to the learner the full gradient of a function at the queried location, instead of only the sign of the gradient as in the present model.

## Footnotes

[1]Note that the query $Q_i$ does not explicitly depend on previous queries, $\{Q_1, \dots, Q_{i-1}\}$, but only their responses. This is without the loss of generality, since for a given value of $Y$ it is easy to see that $\{Q_1, \dots, Q_{i-1}\}$ can be reconstructed once we know their responses and the functions $\phi_1^q, \dots, \phi_n^q$.

[2]This definition of privacy is reminiscent of the error metric used in Probably Approximately Correct (PAC) learning ([14]), if we view the adversary as trying to learn a (trivial) constant function to within an $L_1$ error of $\delta/2$ with a probability great than $1/L$.

[3]Having $\epsilon < \frac{1}{4}\delta$ corresponds to a setting where the learner would like to identify the target with high accuracy, while the adversary is aiming for a coarser estimate; the specific constant $\frac{1}{4}$ is likely an artifact of our analysis and could potentially be improved to being closer to 1. Note that the regime where $\epsilon > \delta$ is arguably much less interesting, because it is not natural to expect the adversary, who is not engaged in the querying process, to have a higher accuracy requirement than the learner. The requirement that $\delta < 1/L$ stems from the following argument. If $\delta > 1/L$, then the adversary can simply draw a point uniformly at random in $[0, 1)$ and be guaranteed that the target will be within $\delta/2$ with a probability greater than $1/L$. Thus, the privacy constraint is automatically violated, and no private learner strategy exists. To obtain a nontrivial problem, we therefore need only to consider the case where $\delta < 1/L$.

[4]We will use the asymptotic notation $f(x) \sim g(x)$ to mean that $f$ is on the order of $g$: $f(x)/g(x) \to 1$ as $x$ approaches a certain limit.

[5]Since all learner strategies considered in the next two subsections will be for the discrete problem, we will refer to them simply as learner strategies when there is no ambiguity.

[6]To avoid the use of rounding in our notation, we will assume that $\delta$ is an integer multiple of $\beta\epsilon$.

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
