[Supplementary Material · supplementary.pdf]

# Supplemental Note for "Query Complexity of Bayesian Private Learning"

Kuang Xu
kuangxu@stanford.edu

## A Figures

---

**Replicated Bisection (Phase 2)**

---

$l^* \leftarrow$ the index of $\mathcal{I}^*$, $K \leftarrow \log\left(\frac{1}{L\epsilon}\right)$, $D_0 \leftarrow \frac{1}{2L}$
**for** $k := 0$ **to** $K - 1$ **do**
    **begin**
    **for** $l := 0$ **to** $L - 1$ **do**
        $Q_{(k+1)L+l} \leftarrow (l-1)\frac{1}{L} + D_k$
    **if** $R_{(k+1)L+l^*} = 0$ (i.e., $X^* > Q_{(k+1)L+l^*}$) **then**
        $D_{k+1} \leftarrow D_k + \frac{1}{L}\left(\frac{1}{2}\right)^k$
    **else**
        $D_{k+1} \leftarrow D_k - \frac{1}{L}\left(\frac{1}{2}\right)^k$
    **end**

---

Figure 1: Pseudo-code for Phase 2 of the Replicated Bisection strategy. $D_k$ represents the distance from a query submitted in the $k$th round to the left end-point of its corresponding sub-interval.

Figure 2: An example of the indices $J(\cdot, X^*)$. Here, $\delta = 0.2$ and $\epsilon = 0.1$, and the target $X^* = 0.15$. The target thus belongs to the first sub-interval in a $\delta$-uniform partition, and the second sub-interval in an $\epsilon$-uniform partition. We have that $J(\delta, X^*) = 1$ and $J(\epsilon, X^*) = 2$.

## B Proofs

### B.1 Proof of Proposition 4.1

*Proof.* Denote by $\tilde{Q}_l$ the query submitted in the $l$th sub-interval during the last round of Phase 2 of the Replicated Bisection strategy. Note that since the positions of the queries relative to their respective sub-interval are identical in each round, we must have that

$$|\tilde{Q}_l - \tilde{Q}_{l'}| \geq \frac{1}{L}, \quad \forall l, l' \subset \{1, \dots, L\}, l \neq l'. \tag{B.1}$$

By the end of the second phase, the adversary knows that the target belongs to the sub-interval $[\tilde{Q}_l - \epsilon, \tilde{Q}_l + \epsilon)$ for some $l \in \{1, \dots, L\}$, but not more than that. Formally, it is not difficult to show that, almost surely, the posterior density of $X^*$ is

$$f_{X^*}(x|(Q_1, \dots, Q_n)) = \frac{1}{2L\epsilon}, \quad \forall x \in \cup_{l=1}^{L}[\tilde{Q}_l - \epsilon, \tilde{Q} + \epsilon), \tag{B.2}$$

and $f_{X^*}(x|(Q_1, \dots, Q_n)) = 0$ everywhere else. Recall that $\epsilon < \delta$ and $\delta < 1/L$ by assumption, we have that

$$\delta/2 < -\epsilon + 1/L, \tag{B.3}$$

where the right-hand side corresponds to the distance between two adjacent intervals $[\tilde{Q}_l - \epsilon, \tilde{Q} + \epsilon)$. In light of Eq. (B.1), this implies that for any interval $G \subset [0, 1)$ with length $\delta$,

$$\mu^{\mathcal{L}}\left(G \cap \left(\cup_{l=1}^{L}[\tilde{Q}_l - \epsilon, \tilde{Q} + \epsilon)\right)\right) \leq 2\epsilon, \quad \text{almost surely,} \tag{B.4}$$

where $\mu^{\mathcal{L}}(\cdot)$ is the Lebesgue measure. Combining the above inequality with Eq. (B.2), we conclude that, for any adversary estimator $\widehat{X}^a$, generated based on $Q$, we have

$$\mathbb{P}\left(|\widehat{X}^a - X^*| \leq \delta/2 \,\big|\, (Q_1, \dots, Q_n)\right) \leq 1 - \frac{L-1}{L} = \frac{1}{L}, \quad \text{almost surely.} \tag{B.5}$$

This shows that the Replicated Bisection strategy is $(\delta, L)$-private. $\qquad\square$

## B.2   Proof of Lemma 5.4

Proof.   Because the random seed, $Y$, is uniformly distributed over $[0, 1)$, the fact that $\sum_{j=1}^{1/\delta} \delta \int_0^1 \xi_{i,y}\, dy \leq \nu$ follows directly from the learner strategy's being $(\epsilon, \nu)$-accurate:

$$\nu \geq \mathbb{P}\left(\widehat{J} \neq J(\epsilon, X^*)\right) = \sum_{j=1}^{1/\delta} \int_0^1 \mathbb{P}(\mathcal{E}_{j,y})\mathbb{P}\left(\widehat{J} \neq J(\epsilon, X^*) \,\big|\, \mathcal{E}_{j,y}\right) dy = \sum_{j=1}^{1/\delta} \delta \int_0^1 \xi_{j,y}\, dy. \tag{B.6}$$

We now show Eq. (5.6). Fix $j \in \{1, \dots, 1/\delta\}$, and $y \in [0, 1)$. We begin by making the simple observation that, conditional on $\mathcal{E}_{j,y}$, the subset of queries $\mathcal{Q}^j$ together with their responses $\mathcal{R}^j$ is sufficient for generating the learner's estimator, $\widehat{J}$, because under this conditioning, any query that lies outside the sub-interval $M_\delta(j)$ provides no additional information about the location of $X^*$ than what is already known. Furthermore, since the random seed $Y$ is fixed to $y$, the $i$th query, $Q_i$, is a deterministic function of the first $i-1$ responses. We conclude that the set of responses $\mathcal{R}^j$ alone is sufficient for generating $\widehat{J}$.

For an event, $\mathcal{E}$, we will denote by $H(A|B, \mathcal{E})$ the conditional entropy $H(A \,|\, B)$ under the probability law $\mathbb{P}(\cdot|\mathcal{E})$:

$$H(A|B, \mathcal{E}) \overset{\triangle}{=} -\sum_{a \in \mathcal{A}, b \in \mathcal{B}} \mathbb{P}(A = a, B = b \,\big|\, \mathcal{E}) \log\left(\mathbb{P}(A = a \,\big|\, B = b, \mathcal{E})\right), \tag{B.7}$$

where $\mathcal{A}$ and $\mathcal{B}$ are the alphabets for random variables $A$ and $B$, respectively. Similarly, define

$$H(A \,\big|\, \mathcal{E}) \overset{\triangle}{=} -\sum_{a \in \mathcal{A}} \mathbb{P}(A = a \,\big|\, \mathcal{E}) \log\left(\mathbb{P}(A = a \,\big|\, \mathcal{E})\right). \tag{B.8}$$

Let $V \in \{0, 1\}^n$ be the vector representation of $\mathcal{R}^j$:

$$V_i = \text{the } i\text{th element of } \mathcal{R}^j, \quad i = 1, 2, \dots, |\mathcal{Q}^j|, \tag{B.9}$$

and $V_i = 1$ for all $i = |\mathcal{Q}^j|, |\mathcal{Q}^j| + 1, \dots, n$. The conditional entropy of $V$ given $\mathcal{E}_{j,y}$ satisfies:

$$
\begin{aligned}
H\left(V \,\big|\, \mathcal{E}_{j,y}\right) &= \sum_{k=1}^{n} H\left(V \,\big|\, \mathcal{E}_{j,y}, |\mathcal{Q}^j| = k\right) \mathbb{P}\left(|\mathcal{Q}^j| = k \,\big|\, \mathcal{E}_{j,y}\right) \\
&\leq \sum_{k=1}^{n} k \mathbb{P}\left(|\mathcal{Q}^j| = k \,\big|\, \mathcal{E}_{j,y}\right) \\
&= \mathbb{E}\left(|\mathcal{Q}^j| \,\big|\, \mathcal{E}_{j,y}\right),
\end{aligned}
\tag{B.10}
$$

where the inequality follows from the fact that, conditional on there being $k$ responses in $\mathcal{R}^j$, we know that only the first $k$ bits of $V$ can be random, and hence the entropy of $V$ cannot exceed $k$, which is the entropy of a length-$k$ vector where each entry is an independent Bernoulli random variable with mean $1/2$. We now invoke the following lemma by Robert Fano (cf. Section 2.1 of [3]).

**Lemma B.1** (Fano's Inequality). *Let $A$ and $B$ be two random variables, where $A$ takes values in a finite set, $\mathcal{A}$. Let $\widehat{A}$ be a discrete random variable taking values in $\mathcal{A}$, such that $\widehat{A} = f(B, C)$, where $f$ is a deterministic function, and $C$ a random variable independent from both $A$ and $B$. Let $p = \mathbb{P}(\widehat{A} \neq A)$. We have that*

$$H\left(A \,\middle|\, B\right) \leq h(p) + p\left(\log |\mathcal{A}| - 1\right), \tag{B.11}$$

*where $H(A \,|\, B)$ is the conditional entropy of $A$ given $B$.*

We apply Fano's inequality with the substitutions: $A \leftarrow J(\epsilon, X^*)$, $B \leftarrow V$, and $\widehat{A} \leftarrow \widehat{J}$. Eq. (B.11) yields

$$H\left(J(\epsilon, X^*) \,\middle|\, V, \mathcal{E}_{j,y}\right) \leq h(\xi_{j,y}) + \xi_{j,y} \log(\delta/\epsilon), \tag{B.12}$$

where we have used the fact that, conditional on the event $\mathcal{E}_{j,y}$, the index $J(\epsilon, X^*)$ can take at most $\delta/\epsilon$ values. By the chain rule of conditional entropy, we have that

$$
\begin{aligned}
H\left(J(\epsilon, X^*) \,\middle|\, V, \mathcal{E}_{j,y}\right) =& H\left(J(\epsilon, X^*), V \,\middle|\, \mathcal{E}_{j,y}\right) - H\left(V \,\middle|\, \mathcal{E}_{j,y}\right) \\
\geq & H\left(J(\epsilon, X^*) \,\middle|\, \mathcal{E}_{j,y}\right) - H\left(V \,\middle|\, \mathcal{E}_{j,y}\right) \\
\overset{(a)}{=} & \log(\delta/\epsilon) - H\left(V \,\middle|\, \mathcal{E}_{j,y}\right) \\
\overset{(b)}{\geq} & \log(\delta/\epsilon) - \mathbb{E}\left(|\mathcal{Q}^j| \,\middle|\, \mathcal{E}_{j,y}\right),
\end{aligned} \tag{B.13}
$$

where step $(a)$ follows from the fact that conditional on $\mathcal{E}_{j,y}$, $J(\epsilon, X^*)$ is uniformly distributed over $\delta/\epsilon$ possible values, and step $(b)$ follows from Eq. (B.10). Combining Eqs. (B.12) and (B.13) yields

$$
\begin{aligned}
\mathbb{E}\left(|\mathcal{Q}^j| \,\middle|\, \mathcal{E}_{j,y}\right) \geq & \log(\delta/\epsilon) - H\left(J(\epsilon, X^*) \,\middle|\, V, \mathcal{E}_{j,y}\right) \\
\geq & \log(\delta/\epsilon) - \left(h(\xi_{j,y}) + \xi_{j,y} \log(\delta/\epsilon)\right) \\
= & (1 - \xi_{j,y}) \log(\delta/\epsilon) - h(\xi_{j,y}).
\end{aligned} \tag{B.14}
$$

This proves Lemma 5.4. $\qquad\qquad\square$.

## B.3 Proof of Proposition 5.6

Proof. Fix $n \in \mathbb{N}$ and a continuous learner strategy, $\phi \in \Phi_n$, such that $\phi$ is both $\epsilon$-accurate and $(\delta, L)$-private. Let $\widehat{X}$ the estimator of $\phi$. It suffices to show that there exists a function $f : [0, 1) \to \mathcal{M}_{\beta\epsilon}$, such that by using the same queries as $\phi$, and setting $\widehat{J} = f(\widehat{X})$ we obtain a $(\beta\epsilon, \beta^{-1})$-accurate and $(\delta, L)$-private discrete learner strategy. Specifically, let $\phi^D$ be the discrete learner strategy that submits the same queries as $\phi$, and produces the estimator

$$\widehat{J} = J(\beta\epsilon, \widehat{X}). \tag{B.15}$$

That is, $\widehat{J}$ reports the index of the sub-interval in the $\beta\epsilon$-uniform partition that contains the continuous estimator, $\widehat{X}$.

We first show that the induced discrete learner strategy is $(\beta\epsilon, \beta^{-1})$-private. The intuition is that if the target $X^*$ is sufficiently far away from the edges of the sub-interval in the $(\beta\epsilon)$-uniform partition to which it belongs, then both $X^*$ and $\widehat{X}$ will belong to the same sub-interval, and we will have $J(\beta\epsilon, \widehat{X}) = J(\beta\epsilon, X^*)$. To make this precise, denote by $\mathcal{G}_{\beta\epsilon}$ the set of end points of the sub-intervals in the $(\beta\epsilon)$-uniform partition: $\mathcal{G}_{\beta\epsilon} \overset{\triangle}{=} \{0, \beta\epsilon, 2\beta\epsilon, \dots, 1 - \beta\epsilon, 1\}$. Let $\mathcal{S}$ be the set of all points in $[0, 1)$ whose distance to $\mathcal{G}_{\beta\epsilon}$ is greater than $\epsilon/2$:

$$\mathcal{S} = \{x \in [0, 1) : \min_{y \in \mathcal{G}_{\beta\epsilon}} |x - y| > \epsilon/2\}. \tag{B.16}$$

It is not difficult to show that the Lebesgue measure of $\mathcal{S}$ satisfies $\mu^{\mathcal{L}}(\mathcal{S}) = \epsilon/(\beta\epsilon) = \beta^{-1}$, where $\epsilon$ is the length of the intersection of $\mathcal{S}$ with each of the $(\beta\epsilon)^{-1}$ sub-intervals in a $(\beta\epsilon)$-partition. Since $\phi$ is $\epsilon$-accurate, we know that $\widehat{X}$ must be no more than $\epsilon/2$ away from $X^*$, and hence $\widehat{J} = J(\beta\epsilon, X^*)$ whenever $X^* \notin \mathcal{S}$, which implies

$$\mathbb{P}\left(\widehat{J} \neq J(\beta\epsilon, X^*)\right) \leq \mathbb{P}\left(X^* \in \mathcal{S}\right) = \mu^{\mathcal{L}}(\mathcal{S}) = \beta^{-1}. \tag{B.17}$$

This shows that $\phi^D$ is $(\beta\epsilon, \beta^{-1})$-accurate.

We next show that $\phi^D$ is also $(\delta, L)$-private. For the sake of contradiction, suppose otherwise. Then, there exists an estimator for the adversary, $\widehat{J}^a$, such that

$$\mathbb{P}\left(\widehat{J}^a = J(\delta, X^*)\right) > 1/L. \tag{B.18}$$

We now use $\widehat{J}^a$ to construct a "good" adversary estimator for the continuous version: let $\widehat{X}^a$ be the mid point of the sub-interval $M_\delta(\widehat{J}^a)$, where $M_\delta(j)$ is the $j$th sub-interval in the $\delta$-uniform partition. If $\widehat{J}^a = J(\delta, X^*)$, then $M_\delta(j)$ contains $X^*$, and since the length of $M_\delta(j)$ is $\delta$, we must have $\left|\widehat{X}^a - X^*\right| \le \delta/2$, and from Eq. (B.18), this implies

$$\mathbb{P}\left(\left|\widehat{X}^a - X^*\right| \le \delta/2\right) > 1/L. \tag{B.19}$$

We therefore conclude that if an estimator satisfying Eq. (B.18) did exist, then the original continuous learner strategy, $\phi$, could not have been $(\delta, \epsilon)$-private, which leads to a contradiction. We have thus shown that $\phi^D$ is $(\beta\epsilon, \beta^{-1})$-accurate and $(\delta, L)$-private. Because $\phi^D$ uses the same sequence of queries as $\phi$, we conclude that $N(\epsilon, \delta, L) \ge N^D(\beta\epsilon, \beta^{-1}, \delta, L)$. This proves Proposition 5.6. $\square$