[Reviews · NeurIPS 2018]

Reviewer 1



This paper studies Bayesian private learning, a new learning problem where the learner can make queries to identify the target, and tries to hide the learned target so that the queries (without responses) leak little information. The paper focus on the Bayesian setting, where the target concept is chosen at random, and the accuracy and privacy guarantees are measured with the randomness of the learner/adversary/target concept. It gives almost matching upper and lower bounds on the query complexity in this setting (Theorem 2.1). Overall, this is a nice paper that studies a new query learning setting, and its technical results appear to be solid. Some comments: 1. Only the threshold learning problem is studied in this paper. It would be much more interesting to consider other concept classes, for example, learning intervals, linear classifiers, etc. It seems that if the concept class has some invariance (in terms of shifting, rotation), then the replicated bisection strategy may be generalizable. 2. In Definition 1.3, can we replace the two privacy parameters (delta, L) with only one? For example, define phi to be delta-private if for any adversary estimator \hat{X}^a, E|\hat{X}^a - X^*| >= \delta. 3. If I understand correctly, this paper only focuses on the case where X^*'s prior is uniform over [0,1]. What happens if one changes the prior in this setting - does the problem become easier or harder? It would be interesting to establish prior-dependent query complexity bounds in this setting. Edit after reading author feedback: This paper has a lot of potential to be extended to more complex query learning settings (e.g. continuum-armed bandits). On the other hand, I agree with Reviewer 5 that the setting studied in this paper is a bit too artificial (only noiseless binary search is considered).

Reviewer 2



Summary: This paper considers a Bayesian private learning problem, where the learner tries to estimate a point in an interval via a sequence of queries (with binary response) while protecting the information about the actual point from an adversary who could observe the queries (not the responses). The main contribution of the paper is deriving a lower bound on the query complexity. They observe that the sample complexity grows linearly with the privacy parameter L (as eps ->0). They also show that the lower bound is, in fact, tight by formally proving an upper bound for an existing algorithm for this problem. For the proof, they use the standard tools in (information theoretic) minimax lower bound analysis, along with a specialized discrete approximation of the original continuous problem. Comments: The paper provides a concrete theoretical statement on Bayesian private learning. The paper is well organized with a clear introduction, motivation, and problem statement. The proofs seem to be technically correct as well. Authors have given clear insights into the capacity of the problem.

Reviewer 3



- Summary & Strength The paper studies a sequential learning with binary feedback with a privacy concern where the learning objective is to locate a latent value with some confidence while hiding it from a spying adversary who observes all the queries but not the responses. The authors obtain a tight fundamental limit of the query complexity using Fano’s inequality, which is a standard tool for the lower bound analysis. The fundamental limit matches the query complexity of an upper bound algorithm (Replicated Bisection strategy) in the asymptotic order which is originally proposed in [11, 15]. The query complexity analysis shows a privacy-efficiency trade-off. The proof seems sound, and the paper is easy to follow. - Weakness & Comment The proposed model is somewhat artificial. In particular, all the responses are assumed to be correct, whereas noisy response is common in practice. Hence, it limits to connect the considering problem to a motivating example in practice. In addition, there is a line of works e.g., [BA 2008, 13], that study the query complexity even under noisy responses (which lead to interesting applications) although there is no privacy concern. However, there is no justification or motivation on the noiseless response. - Minor comments I’m not sure that the notion of the random seed Y is necessary. Line 34: identity function > indicator function Line 44: $R^i := \emptyset$ > $R^1 := \emptyset$ Line 46: “guess’ > “guess” Line 61: any an adversary > any adversary Line 163: $L \log (1/\epsilon)- L (\log L -1)$ > $L \log (1/\epsilon)- L (\log L -1) - 1$ Eq. 5.7: ${\cal E}_{i, j}$ is used without defining it. -- [BA 2008]: Ben-Or, Michael, and Avinatan Hassidim. "The bayesian learner is optimal for noisy binary search (and pretty good for quantum as well)." Foundations of Computer Science, 2008. FOCS'08. IEEE 49th Annual IEEE Symposium on. IEEE, 2008.